# Nucleosome Remodeling at the Yeast *PHO8* and *PHO84* Promoters without the Putatively Essential SWI/SNF Remodeler

**DOI:** 10.3390/ijms24054949

**Published:** 2023-03-03

**Authors:** Corinna Lieleg, Ana Novacic, Sanja Musladin, Andrea Schmid, Gözde Güçlüler Akpinar, Slobodan Barbaric, Philipp Korber

**Affiliations:** 1Biomedical Center (BMC), Molecular Biology, Faculty of Medicine, LMU Munich, Planegg-Martinsried, 82152 Munich, Germany; 2Laboratory of Biochemistry, Faculty of Food Technology and Biotechnology, University of Zagreb, 10000 Zagreb, Croatia

**Keywords:** nucleosome remodeling, *Saccharomyces cerevisiae*, PHO8, PHO84, SWI/SNF, chromatin remodelers

## Abstract

Chromatin remodeling by ATP-dependent remodeling enzymes is crucial for all genomic processes, like transcription or replication. Eukaryotes harbor many remodeler types, and it is unclear why a given chromatin transition requires more or less stringently one or several remodelers. As a classical example, removal of budding yeast *PHO8* and *PHO84* promoter nucleosomes upon physiological gene induction by phosphate starvation essentially requires the SWI/SNF remodeling complex. This dependency on SWI/SNF may indicate specificity in remodeler recruitment, in recognition of nucleosomes as remodeling substrate or in remodeling outcome. By in vivo chromatin analyses of wild type and mutant yeast under various PHO regulon induction conditions, we found that overexpression of the remodeler-recruiting transactivator Pho4 allowed removal of *PHO8* promoter nucleosomes without SWI/SNF. For *PHO84* promoter nucleosome removal in the absence of SWI/SNF, an intranucleosomal Pho4 site, which likely altered the remodeling outcome via factor binding competition, was required in addition to such overexpression. Therefore, an essential remodeler requirement under physiological conditions need not reflect substrate specificity, but may reflect specific recruitment and/or remodeling outcomes.

## 1. Introduction

The packaging of eukaryotic genomes into chromatin, especially into nucleosomes consisting of 147 base pairs (bp) of DNA wrapped around a histone octamer [1,2], is primarily repressive to all processes that require access to DNA, like transcription or replication [3]. Therefore, in contrast to prokaryotes [4], the default state of eukaryotic genomes corresponds to an “off-state” and most genome activities necessitate opening (“remodeling”) of chromatin structure and its nucleosome constituents [5]. This fits to the regulatory requirements of multicellular eukaryotes where only a small fraction of the genome is expressed in each cell. Unicellular eukaryotes, like the budding yeast *Saccharomyces cerevisiae*, are atypical in this regard, as they constitutively express most of their genes. Accordingly, the nucleosome organization at most yeast promoters corresponds to an “open door policy,” [6] where a nuclease hypersensitive site (HSS) (for discussion of the alternative terms nucleosome-depleted region (NDR) versus nucleosome-free region (NFR) see [7,8]) of up to 150 bp length just upstream of the transcription start site allows access to the transcription machinery [5,8,9]. Nonetheless, some genes are repressed under standard growth conditions and become induced upon specific, often environmentally triggered, signaling. The promoters of these inducible genes are often kept in a repressed state by positioned nucleosomes over functional elements, like the TATA and UAS (upstream activating sequence, i.e., yeast enhancers) elements, and nucleosome remodeling is a crucial prerequisite [10] for their activation [11,12].

Promoters of the phosphate response (PHO) genes in *S. cerevisiae* are paradigmatic for this latter type (reviewed in [13,14]). PHO signaling is a nutrient-dependent regulatory pathway that responds to intracellular inorganic phosphate (P_i_) levels [15,16]. The principal transactivator Pho4 activates numerous PHO genes constituting the PHO regulon. Of these, the *PHO5*, *PHO8* and *PHO84* genes serve since many years as models for nucleosome remodeling mechanisms in the course of promoter opening and transcriptional activation upon gene induction and are highly instructive for analogous mechanisms in multicellular eukaryotes. Under conditions of high phosphate supply (+P_i_ conditions), Pho4 is phosphorylated by the Pho80/Pho85 cyclin/cyclin-dependent kinase complex. This phosphorylation prevents gene activation by Pho4 as it expedites nuclear export and prevents nuclear reimport of Pho4, as well as hampers the interaction with Pho4′s cooperative binding partner Pho2 [17,18,19,20,21,22]. Upon P_i_ removal (−P_i_ conditions) and concomitant decrease of intracellular P_i_ levels, Pho80/Pho85 are inhibited by Pho81, and non-phosphorylated Pho4 accumulates in the nucleus, binds to its binding sites, the UASp elements (upstream activating sequence phosphate regulated), triggers promoter chromatin remodeling [23] and transactivates transcription [24]. Accordingly, deletion of *PHO80* or *PHO85* [25] or out-titration of the Pho80/Pho85 kinase activity by *PHO4* overexpression [23] circumvents the actual PHO signalling pathway and leads to PHO gene induction even at +P_i_ conditions.

A long series of studies in many labs, especially regarding the *PHO5* promoter (reviewed in [13]), elucidated the promoter chromatin opening mechanism in exceptional detail. Pho4 recruits via its activation domain [26,27] chromatin modifying enzymes, like the SAGA complex [28,29], that lead to transient hyperacetylation of promoter nucleosome histones [30,31]. Pho4 also recruits, directly or indirectly, ATP-dependent chromatin remodeling enzymes, for example, the SWI/SNF complex [27,29,32]. Such “remodelers” translocate, assemble and disassemble nucleosomes, as well as change their composition with regard to histone variants [33,34,35,36,37], by using the energy of ATP hydrolysis. They are at the core of the promoter-opening mechanism.

One fundamental question regarding remodelers deals with their specificity and essential versus redundant roles. Many remodeling complexes in yeast, like ISW1a, ISW1b, ISW2, INO80, RSC and SWI/SNF, are able to slide nucleosomes along the DNA, but only two of these, RSC and SWI/SNF, are able to disassemble nucleosomes from the DNA [34,36,37]. The RSC complex is the only remodeler essential for viability in yeast [38], while all others can be deleted, even several of them at the same time [39], arguing that their functions are not essential or can be compensated by other remodelers. The SWR1 complex is an example of highly specialized, non-essential but non-compensated function as only this remodeler exchanges canonical histone H2A for the variant histone H2A.Z (Htz1 in yeast) in vivo [40]. Nucleosomes in an *swr1* deletion strain hardly contain any Htz1 despite expression of Htz1 [41].

While it seems clear–although mechanistically still unexplained–that the SWR1 complex exerts a very specific activity that no other remodeler can provide, it is much less clear why a certain remodeling process, like promoter chromatin opening upon induction, should essentially depend on one particular remodeler even though others exert the same or very similar activities, at least in vitro.

To address this question, we turned to the *PHO8* and *PHO84* promoters in budding yeast. At both promoters, there are nucleosomes that cannot be removed in the absence of SWI/SNF activity, even under full physiological induction conditions [42,43,44]. This encompasses all *PHO8* promoter nucleosomes and therefore prevents *PHO8* activation, while it relates to only the so-called “upstream nucleosome” (see below) at the *PHO84* promoter without much effect on full *PHO84* expression. In addition to SWI/SNF, the INO80 complex is also involved in opening both promoters, but not in such an essential way [42,45]. Curiously, even though both promoters depend stringently on SWI/SNF they do not require the RSC complex [46], which would seem an appropriate substitute for SWI/SNF, as both these remodeling complexes belong to the same remodeler family, show very similar activities in terms of nucleosome sliding and disassembly in vitro [34,36,37] and co-operate at other inducible promoters [47,48].

This essential requirement for SWI/SNF during physiological induction of the *PHO8* and *PHO84* promoters is in stark contrast to the remodeler requirements at the *PHO5* promoter [46]. Five different remodelers from all four major remodeler families (SWI/SNF, CHD, ISWI, INO80 families [49]) are redundantly involved in opening *PHO5* promoter chromatin, i.e., the remodelers SWI/SNF, RSC, INO80, Isw1, and Chd1 [31,32,44,45,46,50,51,52].

As all three PHO promoters are coactivated by the same transactivator Pho4, their different dependencies on chromatin cofactors for chromatin opening seemed unlikely due to different recruitment specificities, i.e., in principle, the same set of cofactors should be recruited to each promoter. Instead, it seems possible that there is something particular, maybe linked to DNA sequence or histone modifications, about some nucleosomes that caused the strict SWI/SNF dependency of *PHO8* and *PHO84* promoter nucleosome remodeling in vivo. Conversely, SWI/SNF may possess a mechanistically unique activity. Maybe only this remodeler was able to remodel these particular nucleosomes and/or only SWI/SNF could remove these nucleosomes as remodeling outcome, whereas others would (re-)generate the nucleosome pattern of the repressed state so that on time average, it seemed as if they did not remodel such nucleosomes. Indeed, the chromatin state of the *PHO8* promoter and of the upstream nucleosome at the *PHO84* promoter in *snf2* mutants, where SWI/SNF activity is abolished, always displayed a pattern similar to the repressed state in wt cells [42,43], i.e., other non-SWI/SNF remodelers always regenerated this state after replication.

Mechanistically, the stringent SWI/SNF requirement for remodeling *PHO8* and *PHO84* promoter nucleosomes could reflect at least three different scenarios. First, such particular nucleosomes may need an especially high local remodeling activity and SWI/SNF may be the only remodeler recruited by Pho4 to sufficiently high local levels. Second, only the SWI/SNF complex may efficiently accept such nucleosomes as substrates for remodeling. Third, other remodelers may also remodel such nucleosomes, but only SWI/SNF removes them in the end while the others tend to put them back so that on time average it seems as if they do not remodel such nucleosomes. Thus, the problem could lie in either the local remodeler concentration, in the nucleosome as substrate, or in the remodeler-specific remodeling outcome. In metaphorical terms, the question is if SWI/SNF is the only screwdriver ready at hand to unscrew that screw (recruitment specificity), or if SWI/SNF is the only screwdriver that fits that screw (substrate specificity), or if SWI/SNF is the only screwdriver that will unscrew while other screwdrivers mostly tighten that screw (remodeling outcome specificity). The first scenario should allow SWI/SNF-independent remodeling upon enhanced recruitment of other remodelers, while the second and third scenarios should not allow it. Such strict mechanistic dependency would contrast with the finding that remodelers of different families were all able to remodel nucleosomes of widely different intrinsic stabilities in vitro [53]. Nonetheless, some aspects of nucleosome properties and remodeler functions in vivo may be missed by such in vitro studies. The third scenario would be in line with in vitro demonstrations that different remodelers lead to different remodeling outcomes, e.g., nucleosome sliding, eviction or compositional changes [34,36,37,54] and generate different steady state nucleosome positioning on the same DNA sequence [55,56,57]. Conversely, changing the DNA sequence in a nucleosome can alter remodeling outcomes and remodeler preferences. As a prominent example, the RSC complex prefers remodeling nucleosomes that contain poly(dA:dT) sequences [58] and removes them in a directional way, i.e., in the 5′ direction from the poly(dA) sequence [56]. In retrospect, this explains our earlier observation that introduction of poly(dA:dT) sequences into the “upstream nucleosome” at the *PHO84* promoter relaxed its SWI/SNF requirement or remodeling in vivo [42]. The poly(dA:dT)-containing nucleosome likely became a preferred substrate for RSC, which circumvented the SWI/SNF requirement.

To distinguish these scenarios, we asked if it was possible to circumvent the strict SWI/SNF requirement for nucleosome remodeling at the *PHO8* and *PHO84* promoters. We found two conditions, insertion of an intranucleosomal UASp element or overexpression of *PHO4* that, each on its own, led to *PHO8* promoter nucleosome remodeling in the absence of SWI/SNF activity, while *PHO84* promoter nucleosome remodeling required the combination of both. This argues against substrate specificity but for recruitment strength and/or remodeling outcome specificity underlying the otherwise strict SWI/SNF requirement at both promoters.

## 2. Results

### 2.1. An Additional Intranucleosomal High-Affinity UASp Element Allows PHO8 Promoter Nucleosome Remodeling without SWI/SNF Activity

The *PHO8* gene encodes an vacuolar alkaline phosphatase [59]. The *PHO8* promoter in its repressed state (+P_i_ conditions) is organized into three positioned nucleosomes and three short HSSs (Figure 1A, [60]). The HSS closest to the gene start is just upstream of the TATA box and the other two contain the UASp2 and UASp1 elements, respectively. The consensus sequence of UASp elements is an E-box motif CACGTG and the extent of matching this motif as well as surrounding bases modulates Pho4 binding strength [61,62,63,64]. UASp2 (CACGTGG) is a high-affinity binding site, while UASp1 (CACGCTT) has low affinity and little role in transactivation as its deletion reduces induced levels of Pho8 alkaline phosphatase activity relative to wild type (wt) by only 5% [65]. Upon *PHO8* induction, the −3 nucleosome flanked by the UASp elements becomes remodeled into an extended HSS of about 300 bp, as seen by increased DNaseI or MNase accessibility in indirect end labeling analyses, as well as by increased accessibility of the intranucleosomal HpaI restriction site [60]. This extended HSS and increased HpaI accessibility is not seen in *snf2∆* deletion mutants lacking the Snf2 ATPase subunit of the SWI/SNF remodeling complex or if Snf2 ATPase activity is abolished by a point mutation in the *snf2K798A* allele (Figure 1B, [43]).

To search for conditions that could allow *PHO8* promoter nucleosome removal in the absence of SWI/SNF, we focused on nucleosome −3. “Nucleosome removal” means here any loss of canonical nucleosome structure as monitored by increased accessibility to DNaseI or restriction endonucleases. At the *PHO5* promoter, nucleosome remodeling is aided by the endogenous intranucleosomal UASp2 site, as it contributes binding competition between the high-affinity DNA binder Pho4 and the histone octamer to the remodeling mechanism, like a pry bar [66]. Therefore, we introduced an additional high-affinity UASp element into the −3 nucleosome of the *PHO8* promoter and tested if this allowed removal of this nucleosome in the absence of SWI/SNF. Indeed, the generation of a high-affinity CACGTGG element by three point mutations within the −3 nucleosome (Figure 1A) led to increased HpaI accessibility (54 ± 3% for intranucleosomal high-affinity UASp vs. 9 ± 2% for wt promoter, Figure 1B) and increased DNaseI sensitivity (Figure 1C, HSS between −388 and −874 marker bands, see Figure 2, Figure 3, Figure 4 and Figure 5 for examples of DNaseI patterns with not removed −3 nucleosome) in this region upon –P_i_ induction in *snf2∆* cells, which was similar to the chromatin opening extent of the wt *PHO8* promoter in wt cells (61 ± 8%). This increase depended on the proper UASp sequence as a scrambled, still palindromic non-UASp element (CTCGAGG, Figure 2A, top panel) with the same GC content as the high-affinity UASp at the same position did not have this effect (17 ± 2% HpaI accessibility, Figure 2B).

At the *PHO5* promoter, enhanced remodeling by the intranucleosomal UASp element was mainly due to its intranucleosomal location and not just due to increased Pho4 recruitment potential caused by the presence of an additional high-affinity UASp element [66]. To test whether this was also true at the *PHO8* promoter, we turned its low affinity UASp1 element, which is not intranucleosomal but located in a constitutive HSS (Figure 1A), into a high-affinity (CACGTGG) site (Figure 2A, bottom panel). This also led to a slight increase in HpaI accessibility (29 ± 6%) but hardly any increase in DNaseI sensitivity in the region of the −3 nucleosome upon induction in *snf2∆* cells (Figure 2B). The effect was clearly much less pronounced than for the intranucleosomal high-affinity UASp element (54 ± 3% HpaI accessibility).

We concluded that an additional intranucleosomal high-affinity UASp element allowed circumventing the otherwise strict dependency of *PHO8* promoter nucleosome removal on SWI/SNF activity probably by additional binding competition and less so by increased Pho4 recruitment potential.

### 2.2. PHO4 Overexpression Is Sufficient for PHO8 Promoter Chromatin Remodeling without SWI/SNF

It is well established that the stringency of chromatin cofactor dependency for *PHO5* promoter opening scales reciprocally with induction strength as mediated by the extent of Pho4 recruitment and thereby Pho4 occupancy at the promoter [32]. Therefore, we asked if enhanced Pho4 recruitment alone, beyond physiological levels and without binding competition via an intranucleosomal Pho4 site, could also allow *PHO8* promoter nucleosome removal without SWI/SNF. Elevated *PHO4* expression leads to substantial nuclear levels of non-phosphorylated Pho4 already under +P_i_ conditions and, thereby, to partial PHO induction, resulting in rather extensive chromatin opening at the *PHO5*, *PHO8* and *PHO84* promoters in wt, but not in *snf2* cells (Figure 3, [23,42,45]). Combining *PHO4* overexpression with –P_i_ conditions corresponds to boosted induction beyond physiological “full induction” conditions. Indeed, this boosted induction led to increased HpaI accessibility at the *PHO8* promoter for wt cells compared to mere –P_i_ conditions (HpaI accessibility of 82 ± 3% with vs. 61 ± 8% without *PHO4* overexpression, respectively, Figure 1B and Figure 3). Remarkably, even *snf2* cells showed substantially increased DNaseI sensitivity and HpaI accessibility, consistently somewhat more extensive in the *snf2K798A* than in the *snf2∆* mutant (59 ± 6% vs. 43 ± 9% HpaI accessibility, respectively, Figure 3). This difference between *snf2* alleles was also apparent for the very limited extent of remodeling upon PHO induction without *PHO4* overexpression (HpaI accessibility of 20 ± 1% for *snf2K798A* vs. 9 ± 2% for *snf2∆*, Figure 1B). Chromatin opening upon *PHO4* overexpression at –P_i_ conditions was not significantly increased if it was combined with the intranucleosomal UASp element (HpaI accessibility of 64 ± 18% with vs. 43 ± 9% without intranucleosomal UASp, Figure 1B and Figure 3).

*PHO5* promoter chromatin opening triggered by Pho4 depends on its activation domain and the viral VP16 activation domain is a less potent substitute if fused to the Pho4 DNA binding domain (DBD) [26]. Similarly, we showed here that the *PHO8* promoter nucleosome removal in the absence of SWI/SNF activity was also not possible if just the Pho4 DBD without the Pho4 activation domain was overexpressed (26 ± 1% HpaI accessibility, Figure 4) and was much less extensive upon overexpression of the VP16 activation domain fused to the Pho4 DBD (HpaI accessibility of 38 ± 2% for Pho4DBD-VP16 and less DNaseI hypersensitivity vs. 59 ± 6% HpaI accessibility and full DNaseI hypersensitivity for full length Pho4, Figure 4). As the Pho4 activation domain recruits chromatin cofactors [27], we concluded that increased nuclear Pho4 levels via *PHO4* overexpression led to increased recruitment of other remodeling enzyme(s) that is/are able to remodel *PHO8* promoter nucleosomes in the absence of SWI/SNF.

### 2.3. PHO80 Deletion Is Less Effective Than PHO4 Overexpression Regarding PHO8 Promoter Nucleosome Removal without SWI/SNF

It seemed that *PHO8* promoter nucleosome removal without SWI/SNF required elevated nuclear Pho4 levels. We therefore tested the *pho80∆* deletion allele as another way of modulating nuclear Pho4 levels in vivo. *PHO80* deletion leads to constitutive nuclear accumulation of non-phosphorylated Pho4 [17] such that *PHO5* and *PHO8* promoter chromatin becomes remodeled already at +P_i_ conditions [60,67]. We wondered if the combination of the *pho80* allele with –Pi inducing conditions, leading to increased induction at the *PHO5* promoter [68], would allow *PHO8* promoter opening without SWI/SNF activity. However, this was hardly the case as judged by the DNaseI pattern (Figure 5) and also HpaI accessibility did not increase significantly (20 ± 1% in the corresponding *snf2K798A PHO80* strain (Figure 1B) vs. 25 ± 7% in *snf2K798A pho80∆* (Figure 5)). In addition, combining the *pho80∆* allele with *PHO4* overexpression and –P_i_ induction did not generate more extensive nucleosome removal than without the *pho80∆* allele (HpaI accessibility of 50 ± 1% with *pho80* allele (Figure 5) vs. 59 ± 6% without (Figure 3)).

### 2.4. RSC Rather Hinders Than Helps Circumventing the SWI/SNF Requirement for PHO8 Promoter Nucleosome Removal under Conditions of PHO4 Overexpression

RSC is the other member of the SWI/SNF remodeler family in yeast [49] that cooperates with SWI/SNF in remodeling at the *PHO5* promoter [46] as well as at other highly expressed genes in yeast [47,48], and may provide the other remodeling activity in the absence of SWI/SNF for remodeling of the *PHO8* promoter −3 nucleosome upon *PHO4* overexpression. However, there were also observations that may argue the opposite way. We showed that RSC depletion via a temperature-sensitive sth1td degron allele at 37 °C did not affect *PHO8* promoter opening upon physiological induction [46]. Moreover, RSC ablation via a temperature-sensitive rsc3-ts allele at 37 °C led to some *PHO8* promoter nucleosome removal already under +P_i_ conditions as seen by increased HpaI accessibility (47% in rsc3-ts [69]) and an altered DNaseI cleavage pattern similar to opened chromatin upon –P_i_ induction, even in the absence of Pho4 (rsc3-ts pho4 double mutant, [69]).

We wished to clarify RSC’s role in *PHO8* promoter nucleosome remodeling in the absence of SWI/SNF. RSC ablation on its own (*sth1td* allele) and in the presence of SWI/SNF increased HpaI accessibility at –P_i_ and 37°C conditions to a similar relative extent (86 ± 5% for *sth1td* vs. 71 ± 8% for wt (Figure 6)) as did *PHO4* overexpression in the wt background at –P_i_ and 30 °C (82 ± 3% with (Figure 3) vs. 61 ± 8% without *PHO4* overexpression (Figure 1B)). Therefore, removing RSC had a similar effect as elevated Pho4 levels, which suggested that RSC is not only not needed for chromatin opening at the *PHO8* promoter, at least in the presence of SWI/SNF, but may even counteract opening. Its absence allowed partial remodeling under repressive [69] and an increased remodeling extent under inducing conditions (Figure 6).

To follow up on this, we asked if nucleosome remodeling would proceed to a higher degree in the absence of SWI/SNF if we combined it with RSC removal. We tested *PHO8* promoter opening at the HpaI site in an *snf2∆ sth1td* double mutant with *PHO4* overexpression upon –P_i_ induction and at 37 °C. Under these conditions, the HpaI site in the *snf2∆ sth1td* strain was slightly but significantly more open than in the *snf2∆* strain (58 ± 2% vs. 52 ± 3% (Figure 6)). This argues that RSC is not one of the other remodelers, but rather hindrance than help for remodeling of the *PHO8* promoter −3 nucleosome in the absence of SWI/SNF.

### 2.5. Combining PHO4 Overexpression with an Additional High-Affinity Intranucleosomal UASp Element Is Necessary to Allow Partial PHO84 Promoter Opening without SWI/SNF

The *PHO84* gene encodes a high-affinity P_i_ transporter of the plasma membrane [70]. Its promoter is among the strongest PHO promoters and harbors five UASp elements (UASpA to UASpE, [64]). Similar to the *PHO5* and *PHO8* promoters, it also undergoes extensive chromatin remodeling upon induction by phosphate starvation [42,68]. At +P_i_ conditions, two high-affinity Pho4 sites (UASpC/D) reside in a constitutive short HSS. This HSS is flanked by two well-positioned nucleosomes (the “upstream” and “downstream” nucleosomes, respectively (Figure 7A)) that each harbor a low affinity Pho4 site (UASpB and UASpE, respectively). The TATA box region has an ambiguous nucleosome organization of intermediate accessibility. Upon –P_i_ induction, the complete *PHO84* promoter region, including the upstream and downstream nucleosomes, as well as most of the TATA box region, turns into an extensive HSS of ~500 bp length of virtually full nuclease accessibility. This chromatin remodeling and gene induction is mainly driven by the UASpC/D/E elements with very little contribution of the UASpA/B elements.

As remodeling of the upstream nucleosome at the *PHO84* promoter depends strictly on the SWI/SNF complex [42], similar to *PHO8* promoter nucleosome remodeling [43], we asked if this dependency could be circumvented in a similar manner as we found here for the *PHO8* promoter. We first tried the combination of –P_i_ induction and *PHO4* overexpression. In contrast to the *PHO8* promoter, this did not allow remodeling of the upstream nucleosome in *snf2∆* or *snf2K798A* cells as monitored by HhaI accessibility (4 ± 1% in *snf2∆* and 11 ± 3% in *snf2K798A* cells (Figure 7B)), which probes specifically the SWI/SNF-dependent upstream nucleosome [42], and by DNaseI indirect end labeling (Figure 7B). Second, we turned the intranucleosomal low affinity UASpB element (CACGTTG), located in the “upstream” nucleosome, via a single point mutation into a high affinity site (CACGTGG, Figure 7A). This also did not allow remodeling of the upstream nucleosome in *snf2∆* cells upon overnight phosphate starvation (HhaI accessibility of 8 ± 2 % (Figure 8)).

Only the combination of the high-affinity UASpB site with *PHO4* overexpression and –P_i_ induction led to partial remodeling of the upstream nucleosome in the absence of SWI/SNF, as seen by 31 ± 1% HhaI accessibility and a DNaseI pattern in the region of the upstream nucleosome that resembled the open state (Figure 8, compare with the DNaseI pattern of the open state in wt under –P_i_ conditions in Figure 7B).

We concluded that *PHO84* promoter upstream nucleosome remodeling depended more strictly on SWI/SNF than remodeling at the *PHO8* promoter. Circumventing the SWI/SNF dependency at the *PHO84* promoter required the combination of enhanced Pho4 recruitment and an intranucleosomal high-affinity Pho4 site and even then remained only partial.

## 3. Discussion

Our study contributes to one of the central questions regarding chromatin remodeling mechanisms: why do certain chromatin transitions depend on specific chromatin cofactors in a more or less stringent way? The differential SWI/SNF requirement for the *PHO5*, *PHO8* and *PHO84* promoters served since many years as an example for the difference between relaxed versus essential requirement although these promoters depend on the same transcriptional transactivator Pho4 [13]. Part of this discussion was the assumption that there may be something special about *PHO8* and *PHO84* promoter nucleosomes so that only SWI/SNF and not even the closely related [49] and more abundant [71] RSC remodeling complex, which is still active in *snf2* mutants, could remodel such nucleosomes. This aspect of nucleosome substrate and remodeling specificity in PHO promoter chromatin opening mechanisms has to be re-considered, as we now demonstrate conditions for *PHO8* and *PHO84* promoter nucleosome removal without SWI/SNF in vivo. We summarize these conditions in Table 1.

Regarding the *PHO8* promoter, the so far seemingly essential SWI/SNF requirement for nucleosome removal does not reflect a mechanistic specialty of this particular remodeling enzyme that would exclusively be able to remodel these special nucleosomes in vivo. Instead, we showed via boosted induction strength upon *PHO4* overexpression that the SWI/SNF requirement at this promoter has to join the ranks of the many chromatin cofactors, e.g., Ino80, Gcn5, Asf1, that are involved in PHO promoter chromatin opening to a degree that reciprocally scales with induction strength [32,42,45,72,73]. Remodelers other than SWI/SNF were able to remodel *PHO8* promoter nucleosomes, but seem to be recruited less efficiently or remodel less effectively so that their activity became apparent only upon increased Pho4 levels. At present, we do not know for sure if the other remodeler(s) is/are directly recruited by Pho4, as would be similar to remodeler recruitment by Gcn4 [47], but the strong effects of *PHO4* overexpression and the dependency on the Pho4 activation domain argue in this direction. We also cannot distinguish if this forced recruitment leads to the same amount of local activity of other remodeler(s) that then remodel(s) with the same effectiveness as otherwise SWI/SNF does. We also do not know if the forced recruitment has to bring in (much) more of the other remodeler(s) as its/their specific remodeling activity per remodeler molecule may be lower than that of SWI/SNF. This will be difficult to distinguish, as it depends on accurate determination of local remodeler occupancy in vivo. Monitoring remodeler occupancy at PHO promoters by chromatin immunoprecipitation (ChIP) or other techniques is notoriously unreliable. For example, RSC binding to the *PHO5*, *PHO8* and *PHO84* promoters was detected by anti-Rsc9-ChIP-chip [74] and anti-Sth1-CUT & RUN [75], but not by native anti-Sth1-ChIP-seq [76] or anti-Rsc8-ChEC-seq [77]. Further, one would have to compare occupancies among remodelers, but all these techniques have unknown and difficult-to-calibrate remodeler-specific efficiencies.

Nonetheless, our data clearly show that forced induction strength allows circumventing the so far seemingly essential SWI/SNF requirement for *PHO8* promoter opening. In addition, we also found that the remodeling outcome could be changed for the non-SWI/SNF remodeler(s) by introduction of an intranucleosomal high-affinity UASp, even at physiological induction strength. As only the intranucleosomal UASp and not just an additional UASp in the neighboring linker region showed this effect, this approach unlikely amounts to just another way of increased recruitment of Pho4 and consequently of other remodelers at the promoter. Instead, this likely is another case where binding competition between the specific DNA binding factor Pho4 and the histone octamer potentiates or alters chromatin remodeling outcome, as we showed previously at the *PHO5* promoter [66]. The effect here is even more pronounced as binding competition not only makes a difference between weak versus strong remodeling at the *PHO5*, but between no and almost full opening at the *PHO8* promoter.

At the *PHO84* promoter, the SWI/SNF-dependency for remodeling of the upstream nucleosome probably corresponds to a combination of recruitment and remodeling outcome specificity as only the combination of forced induction and introduction of intranucleosomal high-affinity UASp element led to partial remodeling without SWI/SNF. It may seem surprising that remodeling at the *PHO84* promoter looked rather extensive by DNaseI indirect end labeling but rather partial by restriction enzyme accessibility (Figure 8). This is due to the very limited digestion degrees employed for DNaseI indirect end labeling so that the patterns reflect the most nuclease-sensitive chromatin states while the majority of templates are still undigested in the region of interest (see strong bands in upper part of lanes for all DNaseI patterns). Therefore, DNaseI patterns are good for demonstrating if hypersensitive regions are generated and for showing how they look. However, this technique has to be complemented by a more quantitative assay, like restriction enzyme accessibility, to monitor which fraction of chromatin templates was actually remodeled. Accordingly, this “partially open” *PHO84* promoter upstream nucleosome upon phosphate starvation and *PHO4* overexpression in *snf2∆* cells and in the presence of the intranucleosomal high-affinity UASp likely means that this nucleosome is fully remodeled for some templates, but only in a smaller fraction than in wt cells under full induction conditions.

For sure, remodeling of *PHO8* and *PHO84* promoter nucleosomes without SWI/SNF calls for one or several other remodeling enzyme(s). At the *PHO8* promoter, the RSC complex is unlikely to contribute to the other remodeler(s), but rather, seems to counteract them. RSC opposes *PHO8* promoter opening with or without SWI/SNF, as HpaI accessibility was always higher upon RSC depletion. The effect was not large, but was significant, and may amount to another example of remodeler-specific remodeling outcome. In this sense, and in line with our earlier conclusions [69], we suggest that the outcome of *PHO8* promoter chromatin remodeling by RSC may correspond to the nucleosome positioning pattern of the repressed, whereas the outcome of remodeling by SWI/SNF corresponds to the pattern of the induced state. RSC activity would be dominant at the promoter under +P_i_ conditions, as it is globally much more abundant than SWI/SNF [71]. Upon PHO induction, Pho4 binds at the promoter and recruits SWI/SNF [27,32] and thereby increases local SWI/SNF levels so that SWI/SNF would overcome/outcompete the RSC activity and remodel this nucleosome in its SWI/SNF-specific way [78], leading to the extended hypersensitive site. Accordingly, if RSC is ablated, the remodeling outcome by SWI/SNF is increased under inducing conditions, as shown here (Figure 6), and already, the global SWI/SNF levels without Pho4-mediated local recruitment under +P_i_ conditions led to partial nucleosome remodeling [69]. Conversely, other remodelers in the absence of SWI/SNF would also have to overcome RSC. Indeed, we noticed (Figure 3) that the ATPase-dead version of the SWI/SNF complex (*snf2K798A*) allowed a bit more opening by other remodelers than the complete absence of the SWI/SNF complex (*snf2∆*). This tendency was the other way around at the *PHO5* promoter where chromatin opening was more impaired in the *snf2K798A* than in the *snf2∆* mutant [45]. These slight differences between both SWI/SNF-inactivating alleles at these two promoters may reflect that both SWI/SNF and RSC contain bromodomains [35] and compete for binding to acetylated histones that are present at both promoters during induction [30,31]. Therefore, we suggest that a nonfunctional SWI/SNF complex that still binds to acetylated histones hinders RSC’s positive role during *PHO5* promoter, but hinders RSC’s negative role during *PHO8* promoter opening.

While RSC is no help for *PHO8* promoter opening, we cannot exclude that RSC may have a role at the *PHO84* promoter in the presence of the intranucleosomal high-affinity UASp, as argued above regarding a possible change of remodeling outcome due to the addition of binding competition.

As we showed that also the INO80 complex [42,45] has a role in physiological opening of the *PHO8* and *PHO84* promoters, we speculate that INO80 contributes to the other remodeling activity in the absence of SWI/SNF. Nonetheless, also other remodelers like Isw1 or Chd1 may contribute, as we showed for the *PHO5* promoter [46]. Our finding that SWI/SNF-independent *PHO8* promoter chromatin opening could be achieved either by increased recruitment or by additional binding competition may suggest that the other remodeler(s) differ(s) in the former vs. the latter case. These questions will be addressed in a future study.

In summary, this and future studies that investigate differential chromatin cofactor requirements across genomic loci help to understand the diversification of specificity in recruitment, substrate and remodeling outcome in the evolution of remodeler ATPases and other chromatin cofactors and their more or less essential roles in physiological remodeling pathways.

## 4. Materials and Methods

### 4.1. Strains, Media, Plasmids and Strain Construction

*Saccharomyces cerevisiae* strains used in this study are listed in Table 2. Strains CY397 *ura3* and CY408 *ura3* were derived from CY397 and CY408 after selection on 5-fluoroorotic acid (5-FOA)-containing medium and confirmed for uracil auxotrophy. Strain CY39780 was generated by transformation of CY397 with a linear DNA fragment of the *pho8* locus where a *LEU2* marker gene cassette was inserted into the *PHO80* coding region. Strain CY407 *pho8::KAN* and CY397 *pho8::KAN* were generated by transformation of CY407 or CY397, respectively, with a linear DNA fragment generated by PCR using genomic DNA of strain Y04315 (EUROSCARF strain) as template and the primers 5′-CTTGCTAGCAACAATAGGCG-3′ and 5′-AGGAAGAAGTTGGCTGGTAG-3′. Successful disruption of the genomic *PHO8* ORF was confirmed by Southern blotting as hybridization with a probe hybridizing in the mid-coding region of the *PHO8* gene no longer gave a signal.

For repressive conditions (high phosphate, +P_i_), yeast strains were grown at 30 °C in YPDA medium (YPD with 0.1 g/liter adenine plus 1 g/l KH_2_PO_4_) for strains without and in YNB selection medium supplemented with the required amino acids for strains with plasmids. Phosphate starvation conditions always corresponded to incubation in phosphate-free synthetic medium with or without lack of amino acids for plasmid selection [45,80]. For transfer to phosphate-free medium, cells were grown and washed in water and resuspended in the phosphate-free medium. If not indicated otherwise, phosphate starvation was overnight.

For results at 37 °C in Figure 6, all strains were grown in YNB medium supplemented with 1 g/l KH_2_PO_4_ (YNBP) either lacking uracil for strains with the *sth1^td^* allele or without selection for all other strains. All strains were first grown to logarithmic phase at 24 °C and then shifted to 37 °C and either YNBP or phosphate-free medium prewarmed to 37 °C (without uracil for *sth1^td^* containing strains) over night. For each experiment involving the *sth1^td^* allele, the respective temperature-sensitivity phenotype was confirmed by growth arrest at 37 °C for more than 12 h after shifting back from phosphate-free to +P_i_ medium, i.e., YNBP minus uracil medium.

The Pho4 overexpression plasmid pP4-70L corresponds to YEpP4 (=pP4-70U [26]) but carries the *LEU2* instead of the *URA3* marker [45]. Plasmids pP4-72 and pP4-72V encode the DNA-binding domain of Pho4 only or a fusion of the Pho4 DNA binding domain with the activation domain of VP16, respectively, under control of the *PHO4* promoter, as described in [26]. Plasmids pP8apain intra UASp_high, pP8apain UASp1_high and pP8apain intra UASp_scrambled were generated by QuikChange mutagenesis (Stratagene) of plasmid pP8apain (=pCB/wt(LEU2) derivative with *PHO8* insert as described in [81]), and plasmid pCB84a-Bhi by QuikChange mutagenesis of pCB84 or pCB84Dmut, respectively [42] using mutagenesis primers, as listed in Table 3. The DNA sequence of the mutated *PHO8* and *PHO84* promoter region was confirmed by Sanger sequencing.

### 4.2. Chromatin Analysis

The preparation of yeast nuclei and chromatin analysis of nuclei by DNaseI indirect end labeling and restriction endonuclease accessibility assays were done as previously described [10,46,80,82,83]. In brief, chromatin digested with DNaseI and deproteinized was secondarily cleaved with BglII or SspI for *PHO8* and *PHO84*, respectively. DNA fragments separated on a 1.5% agarose gel (Loening buffer: 40 mM Tris, 12 mM NaOAc, 36.4 mM acetic acid, 1 mM EDTA) were transferred onto a Nylon membrane (Biodyne^®^, PALL Corporation) by Southern blotting and specifically visualized with a probe abutting the respective restriction site. The probe for the chromosomal *PHO84* locus was a PCR product corresponding to bases −1083 to −1428 from the ATG of the *PHO84* ORF, the probe for the *PHO8* locus was a PCR product corresponding to bases +78 to +568 from the ATG of the *PHO8* ORF. To monitor the *PHO84* plasmid locus, secondary cleavage for DNase I indirect end labeling was done with HindIII and the probe for the plasmid locus corresponded to the HindIII-BamHI fragment of pBR322. Probes were labeled with [α-^32^P]dCTP using the kit PrimeIt II (Stratagene). The four marker bands (lane M) were generated by double digests with HindIII and either ClaI, AgeI, ApaI, or BsBI (from top to bottom in the lanes) for *PHO84,* and BglII and either EcoRV, NdeI, HindIII, or SacI (from top to bottom in the lanes) for *PHO8*. Hybridized membranes were exposed to X-ray films (Fuji Super RX). Films were scanned with an Epson Perfection V700 scanner, either in color or gray scale mode. Scans were imported into Adobe Photoshop CS6 and processed by conversion into grayscale format. Sometimes linear level adjustment was applied to the entire image. Figure layout was done with Adobe Illustrator CS6 and Affinity Designer 1.10.5.1342. Sometimes, parts of the image were rearranged or different exposure times of the same blot were combined, as indicated in the figures and figure legend and Appendix A where original blot images are shown (Appendix A). For restriction nuclease accessibility assays, chromatin was incubated with HpaI or HhaI and deproteinized DNA was cleaved with restriction enzymes that frame the HpaI or HhaI cleavage site: BglII/EcoRVfor HpaI at *PHO8* or HindIII for HhaI at *PHO84*. We controlled that the restriction enzyme concentration was not limiting by using two enzyme concentrations that differed at least 2-fold (75 and 150 U for HpaI, 60 and 240 U for HhaI). The resulting DNA fragments were resolved in agarose gels and Southern blotted as for DNaseI mapping. Probes for the *PHO8* and *PHO8* chromosomal locus were the same as for DNase I mapping. To monitor HhaI accessibility at the *PHO84* plasmid locus, BamHI and EcoRV were used for secondary cleavage and a PCR product from −557 to −310 from the ATG of the *PHO84* ORF on the plasmid locus was used as probe. Quantification of the percentage of cleaved DNA was done by PhosphorImager analysis (Fuji FLA3000, Fujifilm imaging plate, BAS-MP) with Aida Image analyzer software v.4.27 (Raytest). The restriction enzyme accessibility [%] was calculated as the quotient [cut/ (cut + uncut)]. Error bars show either the range of two or the standard deviation of more than two biological replicates.

## Figures and Tables

**Figure 1 ijms-24-04949-f001:**
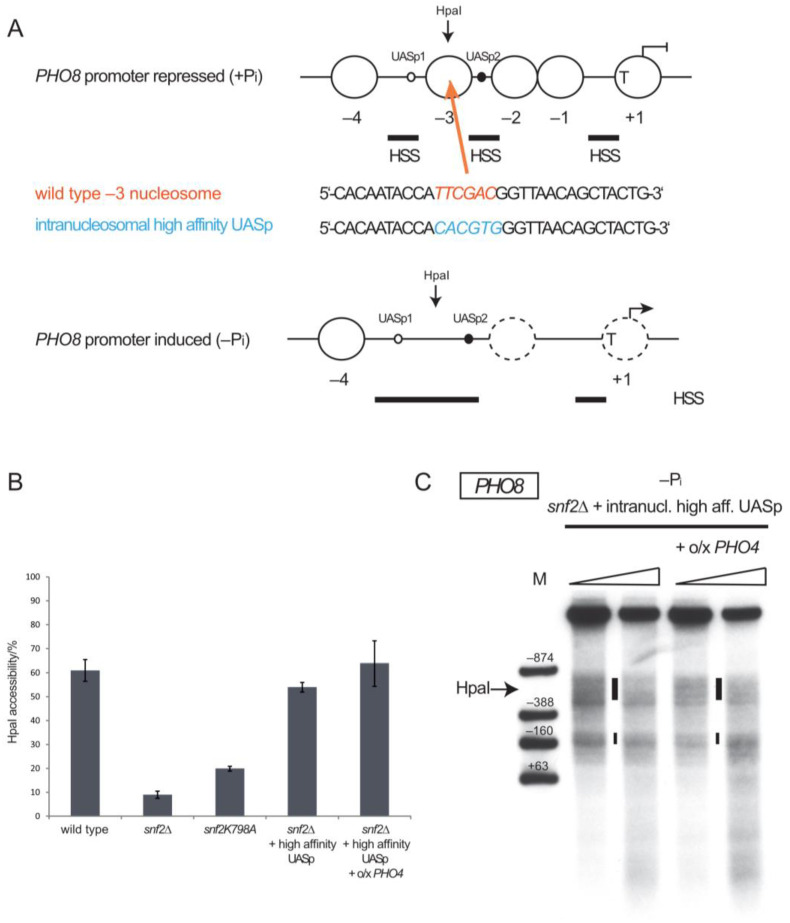
Introduction of an additional intranucleosomal high-affinity Pho4 binding site (UASp) allows nucleosome remodeling at the *PHO8* promoter in the absence of SWI/SNF activity. (**A**) Schematics, approximately to scale, of the nucleosome organization at the *PHO8* promoter in the repressed and induced state (broken arrows show direction and start of transcription in the repressed (blunt arrow) or induced (pointed arrow) state, respectively). Large open circles denote positioned nucleosomes, numbered relatively to transcription start. Horizontal bars represent hypersensitive sites (HSS), dashed circles ambiguous nucleosome organization, small circles high (closed small circles)- or low (open small circles)-affinity UASp and “T” the TATA box position. The position of the HpaI site used for restriction enzyme accessibility assays is indicated as well the position (colored arrow) where the new high-affinity UASp element was introduced in the −3 nucleosome on a plasmid according to the indicated sequence changes. (**B**) HpaI accessibility values for the indicated genotypes, all after overnight incubation in phosphate-free medium (−P_i_). “high affinity UASp” stands for the newly introduced intranucleosomal UASp in the plasmid locus shown in panel A. “o/x *PHO4*” stands for *PHO4* overexpression. Average values and error bars (standard deviation) derived from two or more biological replicates are shown. (**C**) DNaseI indirect end labeling for the *PHO8* plasmid locus and indicated genotypes, all after overnight incubation in phosphate-free medium (−P_i_). Ramps on top of the lanes denote increasing DNaseI concentration. Vertical bars in-between lanes indicate hypersensitive regions. Marker fragments (lane M) were generated by double digests with EcoRV (−874 relative to *PHO8* ATG) or NdeI (−388) or HindIII (−160) or SacI (+61) each with BglII (+571). The approximate position of the HpaI site is indicated.

**Figure 2 ijms-24-04949-f002:**
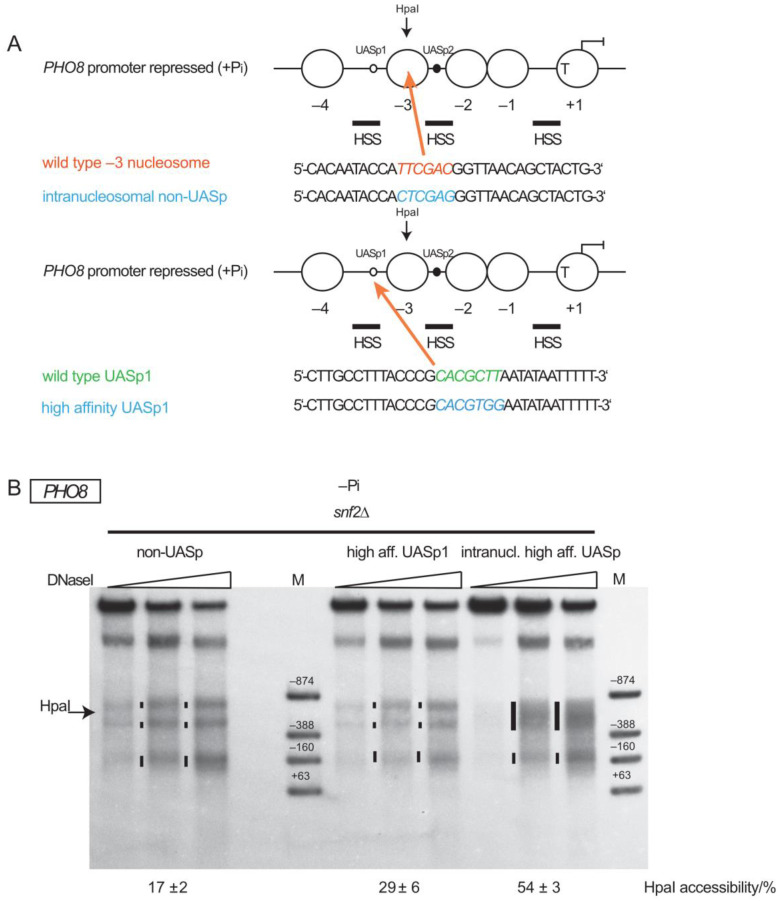
Neither introduction of an intranucleosomal scrambled non-UASp site nor of an additional internucleosomal high-affinity UASp site allowed *PHO8* promoter opening without SWI/SNF activity. (**A**) Schematics analogous to Figure 1A showing the sequence changes for introduction of the scrambled non-UASp site in the −3 nucleosome (top) and the conversion of the low affinity UASp1 site into a high-affinity site (bottom) in the plasmid locus. (**B**) DNaseI indirect end labeling for the *PHO8* plasmid locus analogous to Figure 1C for the indicated genotypes and growth condition (−P_i_). Corresponding HpaI accessibility values analogous to Figure 1B are given below the blots. The condition −Pi *snf2∆* intranucl. high aff. UASp is a biological replicate of the same condition in Figure 1C.

**Figure 3 ijms-24-04949-f003:**
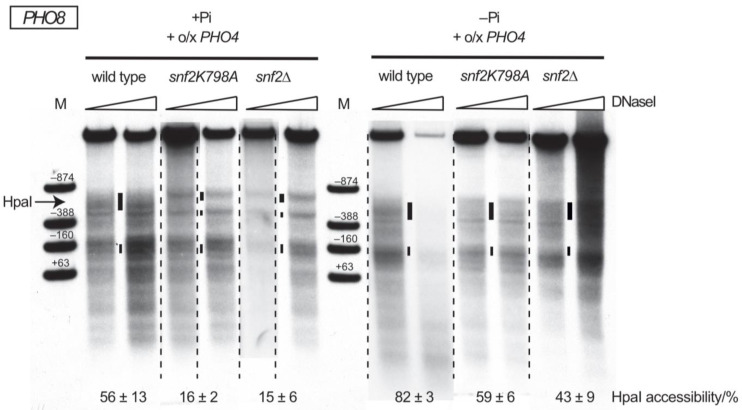
Overexpression of *PHO4* allowed opening of the *PHO8* promoter in the absence of SWI/SNF activity. DNaseI indirect end labeling analysis analogous to Figure 1C for the indicated genotypes and either logarithmic growth in phosphate-containing (+P_i_) or overnight incubation in phosphate-free (−P_i_) medium. Corresponding HpaI accessibility values analogous to Figure 1B are indicated below the blots. Thin dashed vertical lines indicate that gel lanes were combined from films with different exposure times of the same blot in Adobe Photoshop CS6.

**Figure 4 ijms-24-04949-f004:**
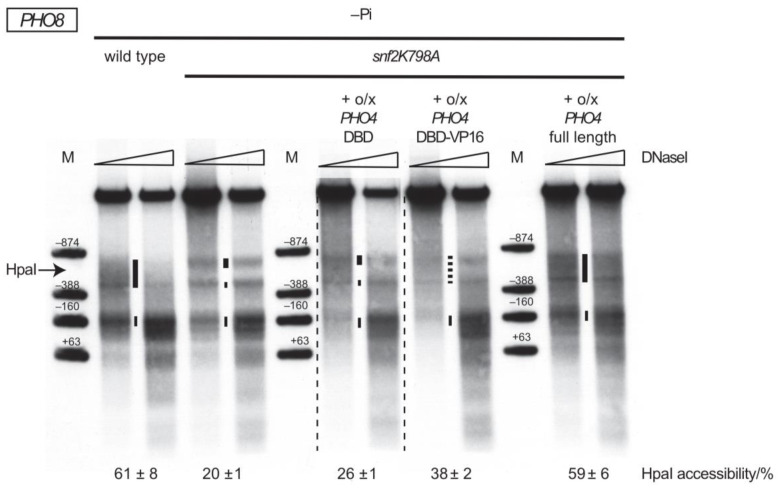
*PHO8* promoter opening in the absence of SWI/SNF activity by *PHO4* overexpression depends on the Pho4 activation domain. DNaseI indirect end labeling and corresponding HpaI accessibility values analogous to Figure 1B,C for the indicated genotypes and growth conditions. Short dashed vertical bar denotes not fully hypersensitive region. Thin dashed vertical lines, as in Figure 3. The condition –Pi *snf2K798A* + o/x *PHO4* full length is a technical replicate of the condition −Pi *snf2K798A* + o/x *PHO4* shown in Figure 3.

**Figure 5 ijms-24-04949-f005:**
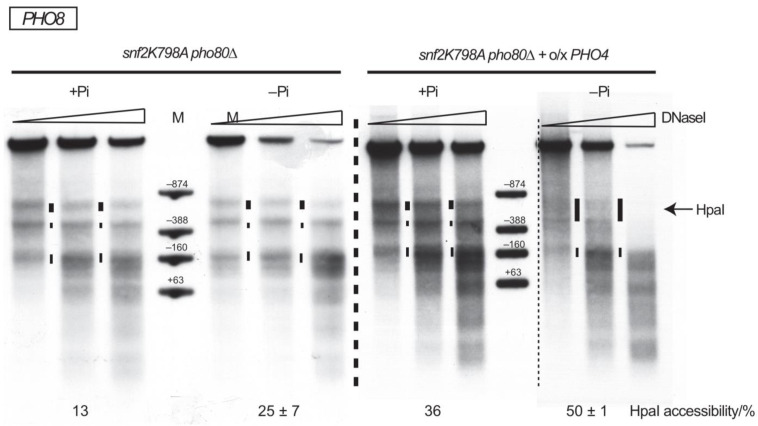
Enhanced induction by *pho80∆* deletion allele only weakly increases *PHO8* promoter opening in the absence of SWI/SNF activity. DNaseI indirect end labeling and corresponding HpaI accessibility values analogous to Figure 1B,C for the indicated genotypes and growth conditions. Thin dashed vertical line, as in Figure 3. Thick dashed vertical line separates samples electrophoresed in different gels.

**Figure 6 ijms-24-04949-f006:**
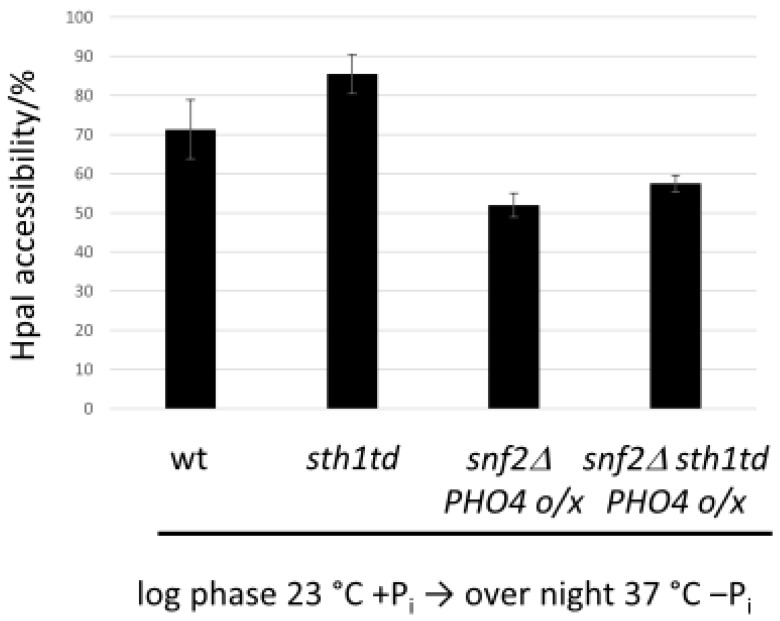
RSC does not help but rather hinders *PHO8* promoter nucleosome remodeling with or without SWI/SNF. HpaI accessibility values analogous to Figure 1B for the indicated genotypes and growth conditions.

**Figure 7 ijms-24-04949-f007:**
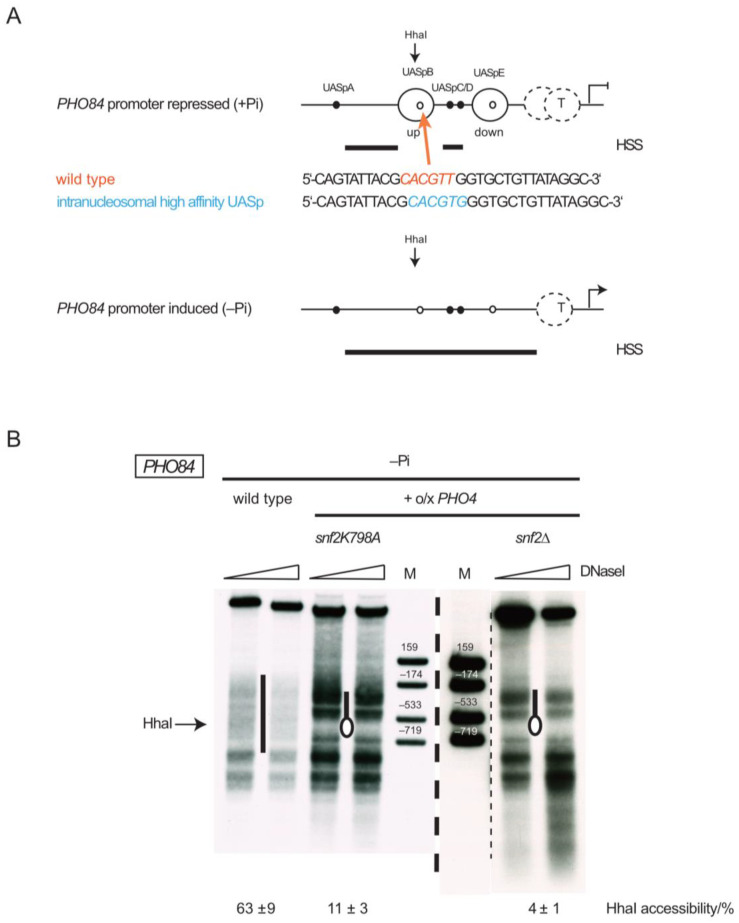
*PHO4* overexpression does not allow remodeling of the upstream nucleosome at the *PHO84* promoter in the absence of SWI/SNF activity. (**A**) Schematics, approximately to scale, of the nucleosome organization at the *PHO84* promoter analogous to Figure 1A. “up” and “down” denotes the nucleosomes “upstream” and “downstream” of the constitutive HSS bearing the UASpC/D elements, respectively. The position of the HhaI site used for restriction enzyme accessibility assays is indicated as well the position (colored arrow) where the new high-affinity UASp element was introduced in the −3 nucleosome on a plasmid according to the indicated sequence changes. (**B**) DNaseI indirect end labeling, as in Figure 2B, but for the chromosomal *PHO84* locus and genotypes, all after overnight incubation in phosphate-free medium (−P_i_). Vertical bars next to lanes indicate hypersensitive regions, the oval the positioned upstream nucleosome at the *PHO84* promoter. Marker fragments (lane M) were generated by double digests with HindIII (−1451) and either ClaI (+159) or AgeI (−174) or ApaI (−533) or BsrBI (−719). The approximate position of the HhaI cleavage site is indicated. Thick dashed vertical line, as in Figure 5. Thin dashed line separates lanes of the same gel but horizontally moved together after cutting away intervening lanes in Affinity Designer 1.10.5.1342.

**Figure 8 ijms-24-04949-f008:**
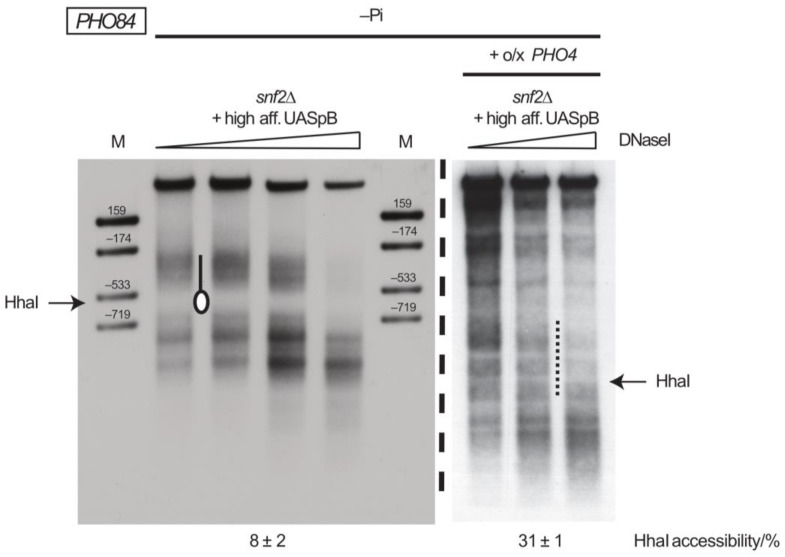
The combination of *PHO4* overexpression with introduction of an additional intranucleosomal high-affinity Pho4 binding site (high-affinity UASpB) allows partial remodeling of the upstream nucleosome at the *PHO84* promoter in the absence of SWI/SNF activity. DNaseI indirect end labeling for the plasmid *PHO84* locus in *snf2∆* cells (CY407) as in Figure 7B after overnight incubation in phosphate-free medium (−P_i_). “+ high aff. UASpB” corresponds to plasmid pCB84a-Bhi, where UASpB is mutated to a high-affinity UASp element according to Figure 7A. No marker for the plasmid locus was included in the gel on the right hand side, but the position of the HhaI site was estimated according to similar published gels (Figure 3B,C in ref. [42]). Even without exact calibration of band gel positions it is clear that the protected region corresponding to the upstream nucleosome (see left hand side of figure) is not detected in the gel on the right hand side.

**Table 1 ijms-24-04949-t001:** Summary of main results.

Relevant Cis Promoter Features	Relevant Trans Factors	Nucleosome Remodeling upon–P_i_ Induction?	
		*PHO8* Promoter	*PHO84* Promoter Upstream Nucl.
wt	wt	yes	yes
wt	no SWI/SNF	no	no
intranucleosomal high-affinity UASp	no SWI/SNF	yes	no
wt	no SWI/SNF *PHO4* overexpression	yes	no
wt	no SWI/SNF *PHO4* overexpression depleted RSC	yes	n.d.
wt	no SWI/SNF *PHO4*-DBD overexpression	no	n.d.
high-affinity UASp1	no SWI/SNF	no	n.a.
wt	no SWI/SNF, no Pho80	no	n.d.
wt	no SWI/SNF, no Pho80 *PHO4* overexpression	yes	n.d.
intranucleosomal high-affinity UASp	no SWI/SNF *PHO4* overexpression	yes	yes

Abbreviations: “nucl.”: nucleosome, “*PHO4*-DBD”: DNA binding domain of Pho4, “n.d.”: not determined; “n.a.”: not applicable.

**Table 2 ijms-24-04949-t002:** Strains used in this study.

Strain Name	Genotype	Short Hand	Source
CY337	*MATa ura3-52 lys2-801 ade2-101 leu2-Δ1 his3-Δ200*	CY wild type	[79]
CY338	CY337 *pho4::URA3*	*pho4∆*	[42]
CY397	*MATα swi2∆::HIS3 swi2(K798A)-*HA-6HIS::*URA3 HO-lacZ*	*snf2K798A*	[79]
CY397 *pho8::KAN*	CY397 *pho8::KanMX4*	*snf2K798A pho8∆*	This study
CY397 *ura*	CY397 *ura3* (after selection on 5-FOA plates)		This study
CY407	CY337 *snf2::HIS3*	*snf2* *∆*	[79]
CY407 *pho8::KAN*	CY407 *pho8::KanMX4*	*snf2* *∆ pho8∆*	This study
CY337 *sth1^td^*	CY337 *sth1Δ::pCUP1-sth1^td^::URA3*	*sth1^td^*	[46]
CY407 *sth1^td^*	CY407 *sth1Δ::pCUP1-sth1^td^::URA3*	*snf2∆* *sth1^td^*	[46]
CY39780	CY397 *pho80::LEU2*	*snf2* *K798A pho80∆*	Hörz group, unpublished
CY408	CY407 *pho4::URA3*	*snf2* *∆ pho4∆*	[30]
CY408 *ura*	CY408 *ura3* (after selection on 5-FOA plates)		This study
BY4741 (=EUROSCARF Y00000)	*MATa his3Δ1 leu2Δ0 met15Δ0 ura3Δ0*	BY wild type	EUROSCARF
Y04315	BY4741 *pho8::KanMX4*	*pho8∆*	EUROSCARF

**Table 3 ijms-24-04949-t003:** Plasmids and primers used in this study.

Plasmid	Source/Mutagenesis Primers	Marker
pP4-70L (*PHO4* o/x)	[45]	*LEU2*
pP4-70U (*PHO4* o/x)	[26]	*URA3*
pP4-72 (*PHO4*-DBD o/x)	[26]	*URA3*
pP4-72V (*PHO4*-DBD-VP16 o/x)	[26]	*URA3*
pP8apain intraUASp_high	this study 5′GTAATCCTAATTTGAGCTCTACACAATACCACACGTGGGTTAACAGCTACTGCA3′ 5′TGCAGTAGCTGTTAACCCACGTGTGGTATTGTGTAGAGCTCAAATTAGGATTAC3′	*LEU2*
pP8apain UASp1_high	this study 5′GATAAGAAGGAAAAATTATATTCCACGTGCGGGTAAAGGCAAGGAAGAATC3′ GATTCTTCCTTGCCTTTACCCGCACGTGGAATATAATTTTTCCTTCTTATC	*LEU2*
pP8apain intraUASp_ scrambled	this study 5′CTCTACACAATACCACTCGAGGGTTAACAGCTACTGC3′ 5′GCAGTAGCTGTTAACCCTCGAGTGGTATTGTGTAGAG3′	*LEU2*
pCB84a-Bhi	this study 5′CAGTATTACGCACGTGGGTGCTGTTATAGGC3′ 5′GCCTATAACAGCACCCACGTGCGTAATACTG3′	*LEU2*

## Data Availability

Not applicable.

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
