# Peer review of "Nucleosome Remodeling at the Yeast *PHO8* and *PHO84* Promoters without the Putatively Essential SWI/SNF Remodeler"

_ijms, 2023, doi:10.3390/ijms24054949_

Round 1
Reviewer 1 Report
The authors have investigated the essentiality of the SWI/SNF in nucleosome Nucleosome remodeling at the yeast PHO8 and PHO84 promoters. The article is very well written. The literature survey is thoroughly done, the experiments are well-designed and the discussion is to the point. I just have one major and one minor question. If the authors can answer both of them then only the manuscript can be accepted.
What is the importance of the present work? To prove that SWI/SNF is not essential, the authors changed a low affinity site to a high affinity site. What is the physiological relevance of this mutation? Has it been observed in happen biologically? If not, then what will be the impact of the present work?
What is loaning buffer?
Reviewer 2 Report
The manuscript ‘Nucleosome remodeling at the yeast PHO8 and PHO84 promoters without the putatively essential SWI/SNF remodeler’ by Lieleg et al. adds good information in the field of SWI/SNF independent chromatin remodeling in yeast.
The topic is notable, the manuscript is well-written and the results are clearly presented. The manuscript can be accepted for publication in its present form.
Reviewer 3 Report
The authors Corrinna Lieleg et al of the manuscript entitled "Nucleosome remodeling at the yeast PHO8 and PHO84 promoters without the putatively essential SWI/SNF remodeler" have done a commendable job of showing that the over expression of the remodeler recruiting transactivator Pho4 removal of PHO8 promoter nucleosomes took place without SWI/SNF chromatin remodelers. They also showed that an addition step of factor binding was required in addition to over expression of PHO4. Additionally, the authors also show that RSC hinders the PHO8 promoter nucleosome removal under conditions of PHO4 over expression.
Some of the major concerns I have with regards to this manuscript are as follows
1. The authors should reconsider revising theta introduction. the introduction is rather too long and should rather be crispy short and focussed. I strongly believe the authors attempted at providing a detailed background for their study, but I believe that keeping it short and to the point and simply highlighting the need for such a study should be sufficient to drive home their point.
2. With respect to the figure 6 it would be good if the text is maininted uniformly across the manuscript. The figure in my opinion seems to be blown up and its reduction in size could be considered.
3. With respect to the figure 8, could you please invert the figure to reduce the background to make it consistent with all the other images presented in the manuscript (Black and white).
4. The figure levels for all the figures could use some additional expiation.
5. The discussion again I believe is particularly long and should be revamped to make it concise and highlighting the important findings. Additionally it would help the readers if a schematic/cartoon explaining the chromatin remodeling in these particular cases can be derived at.
Reviewer 4 Report
Nucleosome remodeling at the yeast PHO8 and PHO84 promoters without the putatively essential SWI/SNF remodeler
In the above manuscript authors tested the possibility of bypassing the essential SWI/SNF chromatin remodelers at the PHO8 and PHO84 chromatin loci upon -Pi starvation. The authors show with DNaseI footprinting and HpaI accessibility assays that with overexpressing Pho4 and/or introducing a intranucleosomal high affinity UASp element this could be partially achieved. The manuscript is well written in general. Although the claims were exclusively made based on DNaseI footprints and HpaI accessibility assays which are semi quantitative, and there are issues and concerns with these blots I chose not to ask for any more experiments. I am ok with accepting the manuscript as it is. However, I would like to register the following concerns for the record.
1. As I mentioned above all claims are supported by gel/blot based semiquantitative assays. Hence, require all controls to be run every time on the same gel for reasonable comparisons. Unfortunately, this is not the case for many figures in the manuscript. For instance,
- Figure 1C: There are no snf2∆ and wt controls included for comparison.
- Figure 2: wild type and snf2∆ controls missing
- Figure 3: Wild type without Pho4 overexpression control in both +Pi and -Pi should have been included. Authors combined lanes from different exposure times in photoshop for display (Lines 347-348). This makes the lanes not comparable.
- Figure 5: snf2K798A control alone and wt control missing. Also, DNaseI footprints seemed prominent in pho80∆ alone without +o/x Pho4 and doesn't correlate well with Hpa1 accessibility% given below.
- Figure 8: The DNaseI foot printing gels for snf2∆ +high aff.UASpB and (+high aff. UASpB + o/x PHO4) are not from the same gel for comparison and missing necessary controls to make comparisons.
2. Authors did not use any statistical testing to verify the significance of their findings wherever possible: Figure 1B and Figure 6 . Figure 1B: may not be possible since they did not have three biological replicates. Figure 6: How many replicates and whether the claims supported by statistics/ what is the statistical significance.
3. Figure 6: missing snf2∆ and sth1td/Pho4 ox controls
4. It would have been more convincing if the authors could have atleast shown enhanced Pho4 recruitment at PHO8 promoter upon introduction of the intranucleosomal high affinity UASp element in vivo such as a ChIP-qPCR. I appreciate that they provide some discussion on that at least.
Round 2
Reviewer 3 Report
With regards to the Comment 4
I was requesting them for additional explanation to the figure legends